# Using surface plasmon resonance, capillary electrophoresis and diffusion-ordered NMR spectroscopy to study drug release kinetics

Alena Libánská [1], Tomáš Špringer[2], Lucie Peštová[2], Kevin Kotalík[1], Rafał Konefał[1], Alice Šimonová [3], Tomáš Křížek[3], Jiří Homola[2], Eva Randárová[1] & Tomáš Etrych [1 ✉]

Nanomedicines, including polymer nanocarriers with controlled drug release, are considered next-generation therapeutics with advanced therapeutic properties and reduced side effects. To develop safe and efficient nanomedicines, it is crucial to precisely determine the drug release kinetics. Herein, we present application of analytical methods, i.e., surface plasmon resonance biosensor technology (SPR), capillary electrophoresis, and [1]H diffusion-ordered nuclear magnetic resonance spectroscopy, which were innovatively applied for drug release determination. The methods were optimised to quantify the pH-triggered release of three structurally different drugs from a polymer carrier. The suitability of these methods for drug release characterisation was evaluated and compared using several parameters including applicability for diverse samples, the biological relevance of the experimental setup, method complexity, and the analysis outcome. The SPR method was the most universal method for the evaluation of diverse drug molecule release allowing continuous observation in the flow-through setting and requiring a small amount of sample.

[1] Institute of Macromolecular Chemistry, Czech Academy of Sciences, Prague, Czech Republic. [2] Institute of Photonics and Electronics, Czech Academy of Sciences, Prague, Czech Republic. [3] Department of Analytical Chemistry, Faculty of Science, Charles University, Prague, Czech Republic. ✉email: etrych@imc.cas.cz

Polymer nanosystems with controlled drug release, i.e. polymer-drug conjugates, enable the targeted drug delivery and release in a spatiotemporally controlled manner, enhancing the therapeutic efficacy of the drugs and reducing their systemic side effects[1]. The spatially-controlled drug delivery is enabled by the increased hydrodynamic size of the nanocarriers allowing their accumulation in tumours or inflamed tissues due to enhanced vascular permeability for macromolecules[2,3] as well as the sufficient stability of polymer-drug linkers during blood circulation[4,5]. Site-specific drug delivery is achieved by stimuli-responsive linkers, e.g. pH-sensitive, enzymatically degradable or reduction-sensitive linkers[6–8], releasing the activated drug in target tissues.

To develop nanocarriers with desirable properties, it is essential to precisely evaluate the drug release kinetics in natural environments, e.g. the bloodstream, extracellular tumorous tissues, inflamed tissues, endosomes and lysosomes. Typically, in vitro models of pH-triggered drug release kinetics are used to predict in vivo behaviour due to reduced cost, time, and labour. The gold standard of drug release methods for polymer nanomedicines is the incubation in aqueous buffers under physiological conditions (usually in the range from pH 7.4 to 5.0) followed by separation of the released drug and spectrophotometric quantification[9–11]. However, this method cannot be applied for the determination of release kinetics of several compounds due to the insufficient separation from the carrier or the weak absorbance spectrum for detection.

Several separation methods, e.g. liquid-liquid extraction, dialysis or high-performance liquid chromatography (HPLC)[12], are widely used for the pH-triggered drug release characterisation but these methods are limited by the specific physicochemical properties of the tested polymer and drug molecules. For instance, the extraction of the drug from aqueous solutions to the organic media requires a difference in hydrophobicity between the drug and the polymeric delivery system, thus the hydrophilic drugs released from water-soluble polymers cannot be separated and detected. Direct analysis of the incubated solution by HPLC has several risks (mainly column clogging and destruction or reduced separation capabilities), which can lead to contamination and damage to the column or machine, therefore, this approach is not recommended. Moreover, the degradation of the released drug during the extraction or low extraction efficiency limits the method application. Furthermore, often irreversible sorption of the released drug with the stationary phase or semipermeable membrane occurs[13,14]. Other techniques such as surface plasmon resonance (SPR) biosensor, nuclear magnetic resonance (NMR) spectroscopy, and capillary electrophoresis (CE) can be potentially used to overcome the limitations of these separation methods to provide additional information about drug release.

The SPR biosensor is one of the most advanced label-free optical biosensors[15], which is widely applied in drug development to characterise bio-molecular interactions[16]. However, to the best of our knowledge, this method has not been used to characterise controlled drug release. There is a report on the use of an SPR biosensor to monitor the dissolution of the polymer matrix containing perphenazine in an aqueous solution[17].

Capillary electrophoresis (CE) is suitable for the determination of the release of ionic and/or ionisable drugs[18–23], including the release from polymer conjugates[24] and is considered the technique of choice for such compounds. Despite the elegant simplicity of this method (no pretreatment), its use is very limited in this scientific field.

Generally, NMR spectroscopy (especially multinuclear 1D and/or 2D NMR) has been applied for drug release and drug degradation as a non-invasive analytical method for certain compounds[25]. Additionally, Diffusion-Ordered Spectroscopy (DOSY) can distinguish the signals of different species in solution and measure their diffusion coefficients ($D$), provided that their signals are resolved in the chemical shift dimension. However, it is mainly used for the confirmation of complexation in drug delivery systems[26] or quantification of the diffusion of micro or nano-sized drug cargo in colloidal and gel systems[27,28]. To our knowledge, [1]H DOSY NMR technique has not been used to evaluate the pH-sensitive release of covalently bound drugs.

In this paper, we pioneered the use of SPR biosensor, [1]H DOSY NMR, and CE methods to characterise the pH-triggered release of covalently bound drugs from the polymer nanocarriers. These methods were applied for several stimuli-sensitive polymer conjugates with three different drugs and their credibility was verified with standard HPLC separation with UV-Vis detection. The polymer conjugates were based on the N-2-(hydroxypropyl) methacrylamide (HPMA) copolymers bearing either dexamethasone (DEX), docetaxel (DTX), or hexyl ester of aminolevulinic acid (HAL) bound via a pH-sensitive hydrazone bond which is biodegradable in the acidic environment of tumours or inflamed tissues[4–6,9,29]. The selected drugs represent different therapeutic areas and the selection was based on several factors, mainly their structure and physicochemical properties. DEX was chosen because it is a widely prescribed medication for various inflammatory diseases and is highly hydrophobic. Similarly, DTX was selected as a hydrophobic chemotherapeutic agent used in the treatment of various malignities. Importantly, instability of the DTX structure caused by the presence of hydrolysable ester bonds highly complicates drug release measurement by conventional HPLC method. Finally, HAL which is currently in clinical trials as a very promising tool for photodynamic therapy of cancer was selected as a representative of hydrophilic and charged molecules without UV-Vis signal.

## Results and discussion

The general biological drawbacks of the low molecular weight drugs, e.g. low concentration in target tissues and the non-specific biodistribution associated with toxicity for healthy tissues, can be overcome by their binding to polymeric drug delivery systems improving their pharmacokinetics and accumulation at the desired site[30–33]. Nevertheless, the potential clinical application of polymer-drug conjugates requires detailed physicochemical characterisation and validation not only in terms of their structure but also the stability in conditions mimicking the bloodstream and drug release at the target site. Several analytical methods have been developed to study the drug release from polymeric drug delivery systems but each method has its shortcomings and specific requirements for the tested system characteristics. Herein, we compared the conventional analytical method based on liquid-liquid extraction of the released drug with subsequent HPLC separation and UV/Vis analysis (hereinafter referred to as the HPLC method) with three analytical methods based on CE, [1]H DOSY NMR and SPR. Several conjugates with structurally different drugs of various hydrophilicity/hydrophobicity, i.e. D-DEX, D-DTX or HAL, bound via pH-sensitive linkers were synthesised to evaluate these methods and hydrophilic HPMA-based polymer-carriers were employed as an example drug delivery system.

**Synthesis and characterisation of copolymer drug conjugates.** The copolymer precursors bearing protected hydrazide groups were synthesised by controlled radical RAFT copolymerisation of HPMA resulting in copolymers with very low dispersity ($Đ = 1.1$). The hydrazide deprotection or the removal of the CTA end group did not change the copolymer dispersity or the molar mass. Two copolymers with different molar masses ($M_w = 42$ kg/

**Table 1 Characteristics of the copolymer precursors used for the conjugation of biotin and the test drugs.**

| Sample | $M_w^1$ (kg/mol) | $Đ^1$ (—) | Hydrazides[2] (mol.%) | $D_h^3$ (nm) |
|---|---|---|---|---|
| P1 | 42 | 1.1 | 5.9 | 8.7 |
| P2 | 20 | 1.1 | 5.3 | 7.3 |
| P3 | 41 | 1.1 | 21.0 | 5.8 |

[1]The molecular weights and dispersity were determined by size exclusion chromatography (SEC) using refractive index (RI) and multi angles light scattering (MALS) detection.
[2]The content of deprotected hydrazides was determined spectroscopically after reaction with TNBSA or by NMR.
[3]The hydrodynamic diameter was determined by dynamic light scattering (DLS) in a phosphate buffer (pH 7.4, 0.1 M).

**Table 2 Characteristics of copolymer conjugates for HPLC and $^1$H DOSY NMR analysis.**

| Sample | $M_w^1$ (kg/mol) | $Đ^1$ (—) | Drug[2;3] (wt.%) | Drug molecules per chain | $D_h^4$ (nm) |
|---|---|---|---|---|---|
| P1-D-DEX | 48 | 1.2 | 9.8 | 9 | 14 |
| P1-D-DTX | 49 | 1.1 | 7.7 | 5 | 11 |
| P3-HAL | 62 | 1.2 | 10.5[3] | 30 | 6 |

[1]The molecular weights and dispersity were determined by SEC using RI and MALS detection.
[2]The bound drug content was determined via HPLC analysis after total hydrolysis of the hydrazone bond.
[3]The HAL content was determined using precolumn derivatisation, HPLC separation and subsequent fluorescent detection.
[4]The hydrodynamic diameter was measured by DLS in a phosphate buffer (pH 7.4, 0.1 M).

mol for P1 and $M_w$ = 20 kg/mol for P2) and hydrodynamic sizes ($D_h$ = 8.7 nm for P1 and $D_h$ = 7.3 nm for P2) were prepared to evaluate the influence of the polymer size on the SPR method (Table 1). The content of hydrazide groups was 5.9 mol.% for P1 and 5.3 mol.% for P2 which was sufficient for the attachment of D-DEX or D-DTX and biotin necessary for binding to the SPR chip. The comonomer ratio in the copolymerisation of P3 was adjusted to obtain a copolymer with a higher hydrazide content (21 mol.%) to attach a sufficient amount of HAL and biotin. Indeed, the increased hydrazide content in the precursor P3 decreased the hydrodynamic size and increased the polymer random coil density. We hypothesise that the increased hydrazide content caused stronger non-covalent interactions between the hydrazide groups.

The copolymer conjugates for HPLC, CE and $^1$H DOSY NMR release kinetics were prepared by the conjugation of D-DEX, D-DTX and HAL, respectively via a pH-sensitive biodegradable hydrazone bond with the polymer precursor P1 or P3 (Table 2). The increase in molar mass of the conjugates corresponded to the amount of drug bound to the copolymers - 9.8 wt.% for P1-D-DEX, 7.7 wt.% for P1-D-DTX and 10.5 wt.% for P3-HAL. The drug conjugation did not significantly change the copolymer conjugate dispersity. Importantly, the attachment of more hydrophobic drugs, D-DEX or D-DTX, led to a significant increase in the hydrodynamic size of the conjugates with respect to the polymer precursor (Table 2). Dynamic aggregates were probably formed due to the hydrophobic interaction in the aqueous solution between the hydrophobic drug molecules and the increased hydrodynamic size could partially modify the biodistribution and blood circulation during in vivo therapy.

The stable binding of the copolymer conjugates to the surface of the SPR chip is necessary for the SPR release kinetics study. Herein, the binding was ensured via strong streptavidin-biotin interaction, therefore, copolymer-biotin conjugates were prepared

by the conjugation of biotin to hydrazide groups of copolymers P1, P2 or P3 via a physiologically stable hydrazide bond and subsequent conjugation of D-DEX or D-DTX or HAL via a biodegradable hydrazone bond. The chemical structures of conjugates P1-B-D-DEX, P1-B-D-DTX and P1-B-HAL are displayed in the Fig. 1. Similarly, the molar masses of biotinylated drug conjugates increased correspondingly with the number of attached molecules, while their dispersity remained low (Table 3). The hydrodynamic size was not affected by the biotin attachment and the increase in the size of the biotinylated conjugates was caused by the attachment of hydrophobic drug derivatives as mentioned above. Biotin was sufficiently bound with more than three molecules per copolymer chain, 6.4 wt.% for P1-B, 5.1 wt.% for P2-B and 3.5 wt.% for P3-B.

The copolymer conjugates with different D-DEX (10.7 wt.% for P2-B-D-DEX$_{10\%}$ and 4.8 wt.% for P2-B-D-DEX$_{5\%}$) content were prepared to examine the influence of drug content on the SPR analysis (Table 3). Importantly, the increase in D-DEX content led to a slight increase in the hydrodynamic size, thus proving partial hydrophobic interaction-based formation of aggregates. Additionally, the copolymers of two different molar masses (P1, P2) were used for the synthesis of the conjugates with 10 wt.% of D-DEX (P1-B-D-DEX, P2-B-D-DEX$_{10\%}$). The increase in the molecular weight of the polymer precursor substantially changed the size of the polymer conjugate. We hypothesise that, in this case, the hydrophobic-based interactions played a significant role in the formation of aggregates as P1-B-D-DEX contains almost three times more D-DEX molecules compared to P2-B-D-DEX$_{10\%}$. Importantly, by prolonging the polymer precursor chain, we increased the hydrodynamic size of the conjugates with comparable wt% of D-DEX more significantly than by doubling the weight content of drug molecules in the conjugates with the same chain length. All the polymer conjugates containing the drug molecules and biotin had similar physicochemical characteristics as the polymers in Table 2, thus the attachment of biotin did not significantly change the properties of polymer conjugates.

**Development of an SPR biosensor-based assay for drug release characterisation.** The SPR biosensor is an effective tool to examine the bio-molecular interactions on a sensor surface[34]. Since this technique has not yet been used for pH-triggered drug release characterisation, this section provides detailed information about the sensor surface preparation and assay optimisation. The golden chip was initially modified with a biotin-binding protein, streptavidin (Supplementary Fig. SI7A) and then, biotinylated drug-containing copolymers were attached via high-affinity biotin-streptavidin binding (see Supplementary Fig. SI7B for immobilisation of D-DEX loaded copolymers). The streptavidin-biotin-based surface chemistry was applied because it provided stable surfaces with a thin copolymer layer (a few tens of nanometres), which was entirely monitored by the SPR biosensor (penetration depth~hundreds of nanometres). The consumption of drug-loaded copolymer was very low ( < 1 μg) and of note, the SPR biosensor continuously monitored the initial preparation of the sensor surface, allowing to vary the experimental conditions and quality control over the prepared surfaces.

The initial test involved the effect of the buffer pH on the sensor surface with the immobilised copolymer with or without D-DEX. The injection of the pH 5 buffer into the detection (P1-B-D-DEX) and reference (P1-B) channels decreased the sensor responses in both channels (Fig. 2A), indicating that the decrease originated from both the release of D-DEX from the immobilised copolymer conjugate P1-B-D-DEX and the changes on the sensor surface with the immobilised copolymer as a reaction to the pH

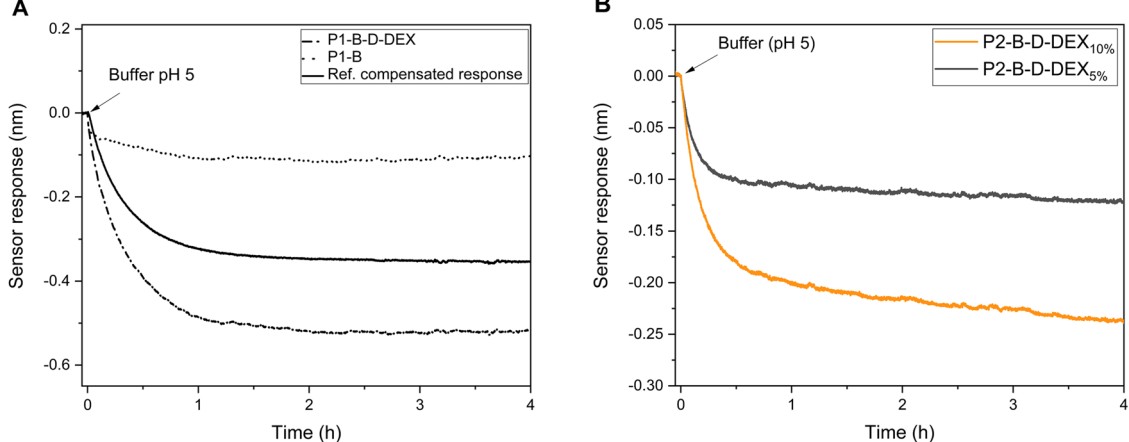

**Fig. 1 Chemical structures of the prepared polymer conjugates.** The different biologically active molecules (R1 = derivate of dexamethasone D-DEX, R2 = derivate of docetaxel D-DTX, R3 = hexyl ester of aminolevulinic acid HAL) are bound via the pH-sensitive hydrazone bond to the HPMA precursor bearing biotin molecule.

**Table 3 Characteristics of prepared samples for SPR analysis.**

| Sample | $M_w^1$ (kg/mol) | $Đ^1$ (—) | Biotin$^2$ (wt.%) | Biotin molecule per chain | Drug$^{3;4}$ (wt.%) | Drug molecules per chain | $D_h^5$ (nm) |
|---|---|---|---|---|---|---|---|
| P1-B | 45 | 1.1 | 6.4 | 8 | - | - | 8.6 |
| P2-B | 22 | 1.1 | 5.1 | 3 | - | - | 7.3 |
| P3-B | 45 | 1.1 | 3.5 | 5 | - | - | 6.0 |
| P1-B-D-DEX | 59 | 1.1 | 6.4 | 8 | 10.8 | 11 | 14.3 |
| P2-B-D-DEX$_{10\%}$ | 21 | 1.1 | 5.1 | 3 | 10.4 | 4 | 10.4 |
| P2-B-D-DEX$_{5\%}$ | 21 | 1.1 | 5.1 | 3 | 4.8 | 2 | 9.9 |
| P1-B-D-DTX | 53 | 1.1 | 6.4 | 8 | 11.9 | 7 | 10.2 |
| P3-B-HAL | 33 | 1.2 | 3.5 | 3 | 15.3$^4$ | 23 | 5.2 |

$^1$The molecular weight and the dispersity were determined using SEC using RI and MALS detection.
$^2$The biotin content was determined via the HABA/Avidin Reagent Kit.
$^3$The bound drug content was determined via HPLC after total hydrolysis of the hydrazone bonds.
$^4$The HAL content was determined using precolumn derivatisation, HPLC separation and subsequent fluorescent detection.
$^5$The hydrodynamic diameter was measured by DLS in a phosphate buffer (pH 7.4, 0.1 M).

**Fig. 2 SPR sensor responses. A** Sensor responses in the detection channel with P1-B-D-DEX (dashed lines) and reference channel with P1-B (dotted line) after the injection of pH 5 buffer. Reference-compensated SPR responses (solid lines) show the D-DEX release from the copolymer. **B** The reference-compensated SPR responses to the D-DEX release from immobilised copolymers P2-B-D-DEX$_{10\%}$ and P2-B-D-DEX$_{5\%}$ in a buffer of pH 5.

change. To distinguish the obtained sensor response induced only by the D-DEX release, the sensor response in the reference channel was subtracted from that in the detection channel. The reference-compensated sensor responses decreased to -0.35 nm for pH 5, respectively. The same procedure was also employed for pH 3 and although non-physiological, it was used to determine the maximum release of drugs from the immobilised copolymer.

The SPR method was further evaluated by measuring the drug release from D-DEX-bound copolymers with varying drug content~5 wt.% (P2-B-D-DEX$_{5\%}$),~10 wt.% (P2-B-D-DEX$_{10\%}$) and molecular weight of copolymer~20 kg/mol (P1-B-D-DEX) and ~60 kg/mol (P2-B-D-DEX$_{10\%}$). In the case of the 5 wt.% and 10 wt.% D-DEX, the release of D-DEX from the copolymer achieved sensor responses of -0.10 and -0.20 nm, respectively

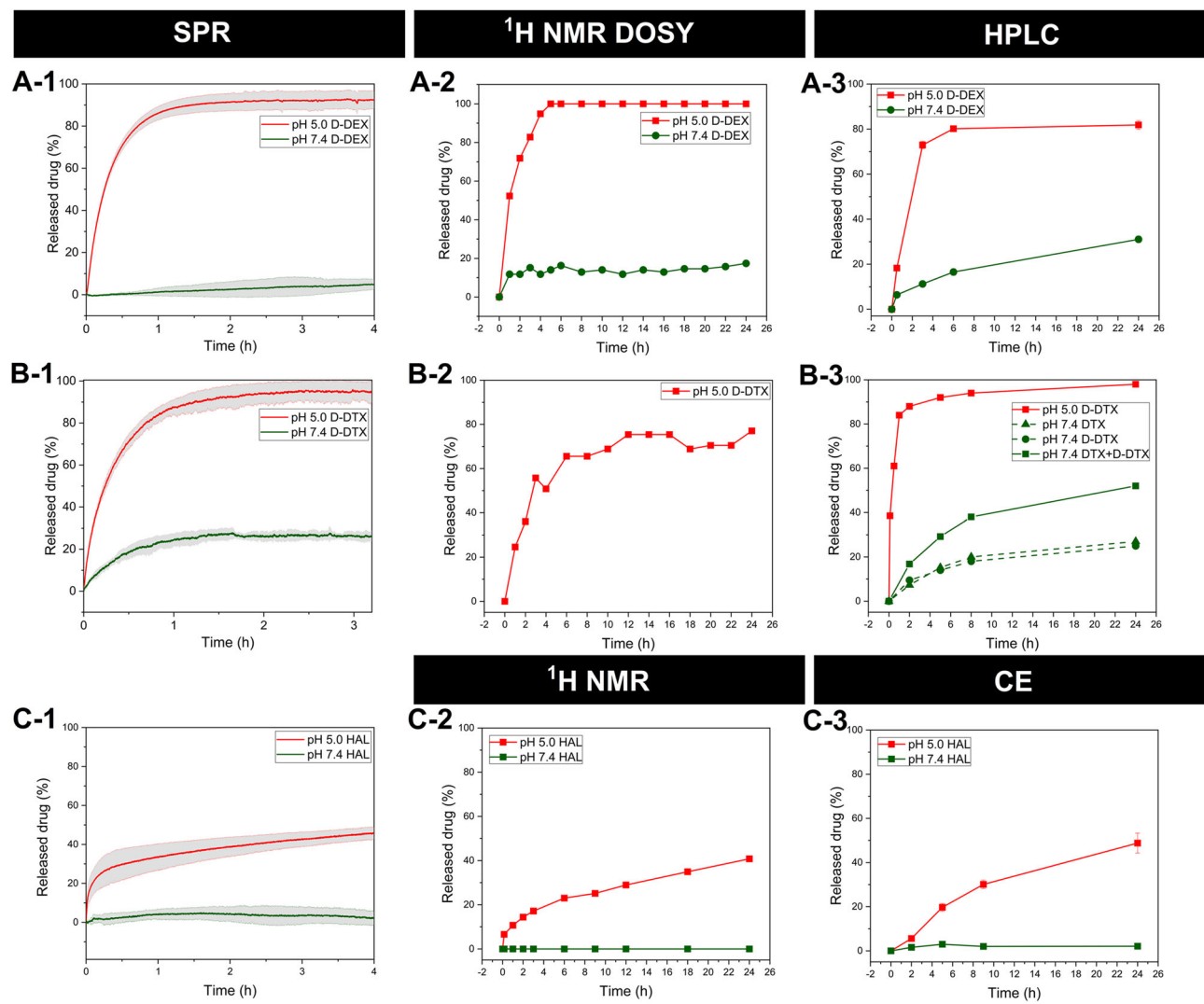

**Fig. 3 In vitro drug release from the polymer conjugates.** D-DEX from P1-B-D-DEX (**A-1**) and P1-D-DEX (**A-2** and **A-3**), D-DTX from P1-B-D-DTX (**B-1**) and P1-D-DTX (**B-2** and **B-3**) and, HAL from P3-B-HAL, (**C-1**) P3-HAL (**C-2** and **C-3**) measured by SPR analysis normalised to the copolymeric precursor (**A-1, B-1, C-1**) with a relative error of ± 5%, ¹H DOSY NMR spectroscopy normalised to the copolymeric precursor (**A-2** and **B-2**) and ¹H NMR spectroscopy (**C-2**) with a relative error of ± 5% followed by HPLC analyses (**A-3** and **B-3**, the relative error is ± 1%) and CE (**C-3**) with a relative error of ± 1%.

(Fig. 2B), which corresponds to the D-DEX content in the copolymers. Due to a low number of biotins in the copolymer chain, the shorter copolymer (P2-B-D-DEX$_{10\%}$, 3 biotin molecules/chain, 20 kg/mol) weakly dissociated from the sensor surface while the longer copolymer (P1-B-D-DEX, 8 biotin molecules/chain, 60 kg/mol) was more stably attached to the sensor surface with the immobilised streptavidin. Thus, the prolongation of the polymer chain beneficially influenced the drifts on the surface of the biosensor allowing better recognition of response associated with the D-DEX release. To avoid any measurement inaccuracies, copolymers were employed with a high drug content and $M_w$ in further experiments to obtain a robust SPR sensor response corresponding to the drug release.

**Release kinetics determined by SPR biosensor**. The developed SPR biosensor was applied to monitor the release of three drugs (D-DEX, D-DTX and HAL) from copolymers at pH 5 (modelling the pH in the lysosomes where the nanomedicines reside after cellular uptake) and 7.4 (modelling the blood conditions). D-DEX was rapidly released from the immobilised copolymer

within the first hour after the pH 5 buffer injection achieving a final release of ~90% of the maximal release obtained for pH 3 (Fig. 3: A-1). The release was very slow for pH 7.4 achieving less than 10% of the maximal release within 4 h. Most D-DTX was released from the immobilised copolymer within the first hour at pH 5 with a final release of nearly 100% (Fig. 3: B-1) within 2 h. At pH 7.4, the release was slow and achieved about 20% of the maximal release within 3 h. Finally, HAL was gradually released (40% within 4 h) at pH 5 with no detectable release observed at pH 7.4 (see Fig. 3: C-1).

Importantly, the SPR biosensor was successfully applied for the characterisation of the release of all three tested drugs (hydrophobic and hydrophilic) showing the method's versatility due to the label-free SPR biosensor monitoring the changes in the refractive index close to the sensor surface produced by the drug molecules released from immobilised copolymers. The real-time SPR biosensor continuously monitored the signal ranging from seconds to several hours providing a continuous characterisation of drug release. Moreover, the flow-through setup is more relevant to biological conditions. Of note, the application of the SPR biosensor simplified the measurement protocol because no

collection and extraction of samples were required. However, pre-analysis modification of the sample is crucial as the binding of biotin to polymer systems is a prerequisite for the SPR chip attachment. Thus, all the polymer conjugates should be enriched by biotin, which requires an additional step for the synthesis of the samples. Noteworthy, the biotin attachment did not cause any changes in physicochemical characterics of the polymer conjugates and did not changed the properties of the buffers used in the experiments. The long-term drug release characterisation (more than several hours) was slightly affected by the interfering effects (e.g. thermal drifts, chemical stability of sensor surface) which decreased the measurement accuracy (see Supplementary Fig. SI8). In addition, the method sensitivity does not allow determining a release lower than 5% within 4 h but this means that the system is sufficiently stable under the given conditions.

**Release kinetics determined by CE.** CE is an analytical method frequently used to quantify hydrophilic ionisable molecules, so we utilised CE with capacitively coupled contactless conductivity detection to determine the release of charged HAL from the hydrophilic polymer conjugate. The release kinetics determined by CE proved the pH-sensitive behaviour of P3-HAL, with up to 50% of HAL released at pH 5 within 24 h and only 3% of HAL released within 24 h at pH 7.4 (Fig. 3: C-3). The exemplary CE electropherogram of P3-HAL after 24 h is displayed in Supplementary Fig. SI9.

This method requires calibration for free drug and optimisation of several measurement parameters, such as background electrolyte (BGE) composition, capillary length and diameter or applied voltage. Conductivity detection allows the determination of compounds with very weak UV absorbance, such as HAL, without the need for derivatisation. CE performed in an open tubular capillary is also relatively tolerant to the sample matrix. If some sample constituents adsorb to the capillary wall, they can be flushed out with a strong base or acid. Also, the number of steps and operator-dependent errors are reduced since no derivatisation or sample pretreatment are required. The procedure is fast, easy to perform and inexpensive due to the need for low sample volumes and BGE for the analysis. However, since the hydrophilicity and ionisability are prerequisites for CE application, the release kinetics of neutral and hydrophobic drugs, i.e. D-DEX and D-DTX, could not be determined by this method.

**Release kinetics determined by NMR spectroscopy.** Firstly, $^1$H NMR spectroscopy was employed for the release kinetics evaluation of D-DEX and D-DTX but due to the low solubility of released drugs in aqueous buffers, the hydrophobic drug signals could not be detected by this method. For comparison, see Supplementary Fig. SI11 where the spectra of P1-D-DEX measured in DMSO$_{-d6}$ and D$_2$O are displayed.

Secondly, the HAL release rate from P3-HAL was evaluated by $^1$H NMR spectra (Supplementary Fig. SI12), with up to 40% of HAL released within 24 h (Fig. 3: C-2) when incubated at pH 5. The release kinetics were calculated from the signal changes at $\delta \approx 4.2$ (signal "1 + 4" overlay of attached and free HAL) and 2.95 ppm (corresponds to free HAL). Surprisingly, the incubation of P3-HAL in D$_2$O buffer at pH 7.4 caused the fission of the signals of bound HAL (at $\delta \approx 4.2$ and 2.78 ppm), which was not observed for free HAL measured under the same conditions. However, there was no signal at $\delta \approx 2.95$ ppm corresponding to free HAL at pH 7.4 (marked red in Supplementary Fig. SI13). We hypothesised that there is no release of HAL from the conjugate and the observed changes in signals at $\delta \approx 4.2$ and 2.78 ppm are related to the charged character of HAL (probably due to gradual change in the charge of amino groups with time at this pH[35]). To

confirm this, P3-HAL was incubated in H$_2$O at pH 7.4 for 24 h, the water was removed by freeze-drying and $^1$H NMR spectrum was measured in DMSO$_{-d6}$, with only negligible differences in the spectra at $t = 0$ h and $t = 24$ h observed proving insignificant HAL release at pH 7.4. Also, 40% of HAL was released within 24 h at pH 5.0 which corresponds to our previous findings. However, this process requires additional steps (including fast freezing and lyophilisation) and a large amount of sample (~5 mg) for one-time point measurement.

In summary, high-resolution $^1$H NMR spectroscopy represents a non-invasive method for determining the release of water-soluble drugs from hydrophilic polymers provided that the signals of the drug and polymer do not overlay and are visible. These prerequisites were fulfilled for the release kinetics of P3-HAL showing comparable results with that observed by CE.

Advanced NMR technique, $^1$H DOSY NMR, is used to characterise the polymer size, presence of monomer impurities and their solution behaviour, or to evaluate the composition of small molecule mixtures, e.g. low molecular drugs. It can distinguish the signals of different species in solution based on their $D$ differences which are directly related to their hydrodynamic size[36]. Herein, we utilised the difference in the $D$ values of the drug-bearing polymer conjugate and the polymer carrier after drug release to quantify the release kinetics by $^1$H DOSY NMR. Thus, the prerequisite parameter for this method is a sufficient difference in the $D$ between the polymer conjugate and polymer carrier.

The highly hydrophobic character of the large molecule D-DEX enabled the quantification of the release kinetics from P1-D-DEX at pH 7.4 and 5 at 37 °C by $^1$H DOSY NMR (Fig. 3: A-2, Supplementary Fig. SI10). The $D$ of P1-D-DEX gradually increased with the incubation time at pH 5 reaching the value of P1 after 5 h, indicating a 100% release of D-DEX. The representative $^1$H DOSY NMR spectra of P1-D-DEX measured directly after sample preparation (blue) and after 24 h at 37 °C (red) at pH 5 are shown in Supplementary Fig. SI14. At pH 7.4, the change of $D$ values was lower, approximately 20% of released D-DEX within 24 h, indicating the relative stability of P1-D-DEX in the blood circulation.

For P1-D-DTX, $^1$H DOSY NMR analysis was only applicable at pH 5 where $D$ gradually increased with incubation time, indicating 80% of released D-DTX after 5 h. Surprisingly, no significant difference in the $D$ of P1-D-DTX and the reference P1 was observed at pH 7.4, therefore the released D-DTX at pH 7.4 could not be quantified. Since the $D$ values are affected by the behaviour of the polymer coil in the solvent, the incubation of P1-D-DTX and P1 in the pH 7.4 buffer may have led to a polymer coil conformation with comparable $D$. The results are summarised in Fig. 3: A-2 and B-2.

No measurable difference in $D$ values of P3-HAL and the reference P3 was observed, which is in line with their comparable hydrodynamic radius (Table 2). Therefore, the quantification of HAL release rates by comparing the $D$ of the polymer-drug conjugate and polymer carrier was not possible. Nevertheless, two-component fitting distinguished the $^1$H DOSY NMR signals originating from P3-HAL, P3 and free HAL at pH 5. However, quantification was not possible as the method cannot characterise the size of individual components, thus only confirming the presence of released HAL at pH 5 (Supplementary Fig. SI15). No signals from free HAL or P3 were observed at pH 7.4, supporting the hypothesis of negligible HAL release under bloodstream-mimicking conditions (see Supplementary Fig. SI16).

The $^1$H DOSY NMR measurements of all the time points were performed in a single tube, thus reducing the amount of sample required and the operator-dependent errors caused by sample handling. The number of tested time points is only limited

by the time needed for each scan and there is no need for sample separation or derivatisation before analysis. However, data evaluation requires a highly experienced operator and a special diffusion probe head with strong field gradient power and optimisation of the acquisition parameters, i.e. diffusion delay, diffusion gradient pulse length and the number of gradient steps are required.

**Release kinetics determined by HPLC.** HPLC with UV-Vis detection is the gold standard method for the determination of drug release from nanomedicines. This multistep method has been used to confirm the pH-sensitive behaviour of the polymer conjugates with D-DEX or D-DTX bound by the hydrazone bond in vitro by incubation in aqueous buffers of different pH[5–7,37]. The pivotal step in this method is the separation of the released drug which requires the quantitative transfer of the released drug from aqueous buffers to the organic phase. Thus, the physico-chemical properties of HAL, especially hydrophilicity and further weak UV absorption and frequent interaction with the stationary phase, hinder the use of the conventional extraction method coupled with HPLC analysis.

The release kinetics of polymer conjugates with D-DEX or D-DTX determined by HPLC proved relative stability of hydrazone bond at pH 7.4 with approximately 30% of released D-DEX and 40% of released D-DTX and DTX in total within 24 h (Fig. 3: A-3 and B-3). The accelerated hydrolysis of the hydrazone bond occurred under acidic conditions, i.e. 80% of released D-DEX and 90% of released D-DTX within 6 h at pH 5. The determination of DTX release was complicated by the drug hydrolysis during the incubation and separation since DTX together with other taxanes is highly unstable at pH 7.4 due to several ester bonds[38,39]. According to the literature, there are five main degradation products, therefore, the calculation via calibration method requires the isolation and characterisation (structure and spectroscopic properties) of the products, followed by calibration. The HPLC chromatogram of the isolated product after the drug release experiment is shown in Supplementary Fig. SI17.

In summary, the HPLC conventional method comprising numerous steps can lead to operator-dependent errors and its applicability requires the distinctive solubility difference between the released drug and remaining polymer and polymer conjugate due to the extraction from the aqueous to organic phase. Moreover, precise calibration of the released drug and its sufficient absorbance is essential, so this technique is not suitable for all drug molecules. In addition, the method does not enable continuous evaluation of drug release, only at preselected time points, so the number of time points determines the amount of sample required for analysis.

**Head-to-head comparison of the methods used for release kinetics.** Each analytical method applied to study the drug release has various shortcomings and specific requirements for the tested system characteristics (see the scheme in Fig. 4 where the overall scheme of the method evaluation approach is displayed). Unlike the well-used HPLC method, the approaches utilising [1]H DOSY NMR, CE with contactless conductivity detection or an SPR biosensor to quantify drug release from polymer conjugates have not been previously reported. Herein, we evaluated the overall suitability of each method for drug release quantification using several parameters determining the properties and advantages of the studied methods (Table 4) including (i) applicability for diverse samples concerning the physicochemical character of the drug molecules and the amount of sample required for analysis; (ii) method complexity and (iii) the analysis outcome.

**Sample properties.** The approach using an SPR chip to quantify the release of hydrophobic, hydrophilic and charged drugs from polymer conjugates proved to be the most universal method for the physicochemical character of various samples. The drug molecules do not need to have highly specific properties such as UV/Vis absorbance or considerable hydrophobicity. In addition, the SPR method directly records the hydrolysis of the hydrazone bond and washing of the released drug from the chip surface, therefore the potential degradation of the released drug, e.g. DTX, should not affect the results. Thus, the SPR method is widely applicable for a variety of structurally different drugs as well as unstable compounds in contrast to the limitation of HPLC, CE or NMR methods. HPLC can only measure molecules with sufficient absorbance, no irreversible interaction with the column solid phase and a high partition coefficient between the aqueous buffer and organic phase, whereas CE is only suitable for ionisable hydrophilic drug molecules. While the release of water-soluble drugs from hydrophilic polymers can be determined by [1]H NMR spectroscopy, the [1]H DOSY NMR method requires a measurable difference between the $D$ of polymer-drug conjugate and polymer carrier, thus only polymer conjugates bearing hydrophobic drugs changing the $D$ of the polymer coil after conjugation or drug content can be evaluated. Furthermore, the drug should not physically interact with polymer chains after its release. The amount of sample needed for HPLC or CE analysis is directly dependent on the number of time points since each time point has to be analysed separately, whereas SPR and both NMR methods require only one sample set with the SPR method being more sensitive, therefore requiring the least amount of sample.

**Method settings.** The method complexity, i.e. the number of steps, defines the probability of operator-dependent errors, so the conventional method utilising HPLC comprising numerous pre-analysis steps has prompted the development of methods based on CE, [1]H DOSY NMR and SPR to simplify drug release kinetics evaluation (Table 4). The CE, [1]H NMR and [1]H DOSY NMR methods enable the direct analysis of the sample solution in the incubation buffer with no pre-analysis. Moreover, the overall measurement in a single NMR tube reduces the sample handling to a minimum. The pre-analysis modification necessary for SPR analysis is an additional step requiring suitable functional groups on the polymer chain for biotin attachment. In contrast, the conventional HPLC approach requires the separate incubation of the sample for each time point (due to possible aggregation of released hydrophobic molecules), subsequent extraction, eva-poration and re-dissolution before analysis.

SPR analysis is the only technique to allow the study of the drug release kinetics in a flow-through system with constant buffer flow and continuous detection. Such an experimental setup is more clinically relevant since the polymer-drug conjugates are exposed in vivo to the blood flow. Importantly, since NMR measurements were performed in a single closed tube, the number of time points is only limited by the time needed for each scan, therefore the release measurement is almost continuous. The closed system with discontinuous detection is characteristic of HPLC and CE analysis. However, CE requires no pre-analysis separation and the very low injection volume of the tested sample enables more frequent time points compared to HPLC which requires the sample to be incubated separately for each time point. The method setting for HPLC, CE or both NMR analyses enables unlimited long-term monitoring, while the interfering effects, like thermal drifts during SPR analysis, enable the monitoring for only several hours. However, monitoring the release rate in the order of a few hours is sufficient, as the release

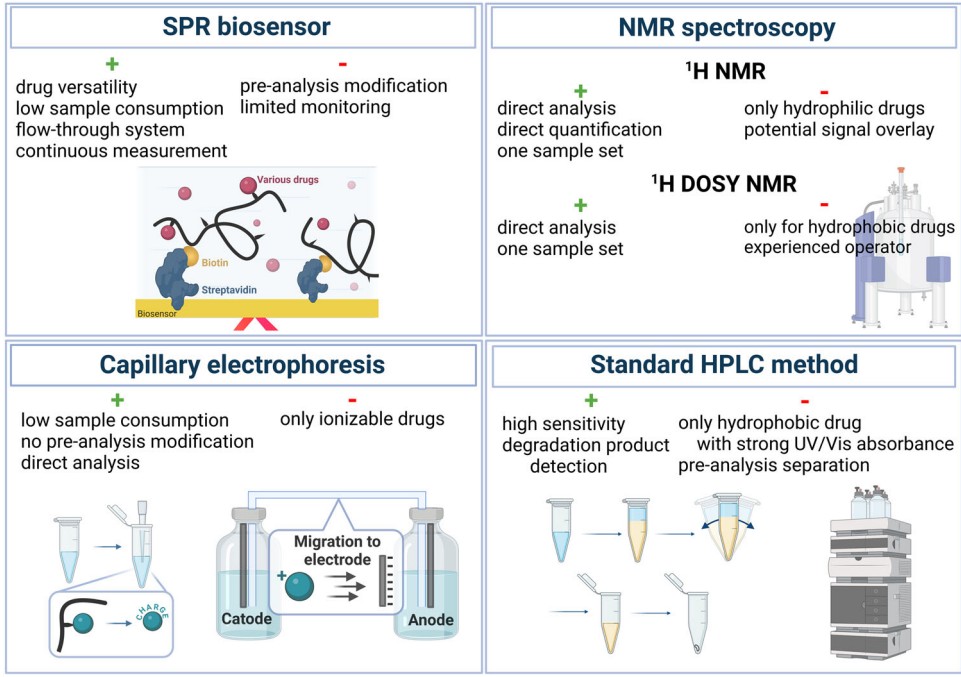

**Fig. 4 Overall scheme of the method evaluation approach.** Each part of the figure is indicating the main positive and negative key parameters for evaluation of SPR biosensor method, NMR spectroscopy, CE and standard extraction method followed with HPLC analysis.

**Table 4 Summary of the benefits and drawbacks of the studied methods.**

| Evaluation parameters | | SPR | CE | NMR | | HPLC |
|---|---|---|---|---|---|---|
| | | | | **¹H** | **¹H DOSY** | |
| Sample | Drug properties | Ho/Hi, charged, unstable | Charged | Hi | Ho | Ho/Hi with UV/Vis absorbance |
| | Sample amount for analysis | ≤1 µg | 2 mg | 3−5 mg | | 3−5 mg |
| Method setting | Pre-analysis process steps | Pre-analysis modification | No | No | | Pre-analysis separation |
| | Flow-through system | Yes | No | No | | No |
| | Examination in time | Fully continuous | Discontinuous | Semi-continuous | | Discontinuous |
| | Long-term monitoring ≥24 h | No | Yes | Yes | | Yes |
| Analysis outcome | Quantification | Reference - polymer precursor | Drug calibration | Direct quantification | Reference - polymer precursor | Drug calibration |
| | Result reproducibility | D-DEX: (a) pH 5, 4 h (b) pH 7.4, 4 h (a) 100% (b) 6% | (a) n.d. (b) n.d | (a) 100% (b) 10% | | (a) 80% (b) 10% |
| | | D-DTX: (a) pH 5, 4 h (b) pH 7.4, 4 h (a) 100% (b) 20% | (a) n.d. (b) n.d | (a) 100% (b) n.d. | | (a) 100% (b) 20% |
| | | HAL: (a) pH 5, 4 h (b) pH 7.4, 4 h (a) 40% (b) 0% | (a) 20% (b) 0% | (a) 20% (b) 0% | | (a) n.d. (b) n.d. |

*Ho* hydrophobic, *Hi* hydrophilic, *n.d.* not determined

rate e slope itself is an important parameter that sufficiently determines the system behaviour in given environments.

**Analysis outcome.** The result of the analysis and its accurate interpretation is the most crucial parameter for any analytical quantification method. The experiment setup, i.e. the flow-through incubation or the frequency of detection time points, significantly influences the outcome of the kinetics study and should be considered during the data interpretation. The high sensitivity of UV/Vis or conductivity detection accompanied by necessary thorough calibration and optimisation of several measurement parameters enables precise quantification of the

released drug by HPLC and CE analysis, respectively, with the detection limit of~1% of the released drug. The direct quantification of the release of hydrophilic drugs drug can also be obtained by ¹H NMR spectroscopy, however, the method sensitivity is lower compared to HPLC and CE analysis, with a detection limit of~5%. In contrast, the quantification by ¹H DOSY NMR and SPR requires the reference measurement of the drug-free polymer carrier and optimisation of complex acquisition parameters, therefore, data interpretation should be performed by an experienced operator. In addition, the sensitivity of ¹H DOSY NMR and SPR is restricted to detect at least~5% of the released drug.

Finally, the reproducibility of the methods' results was analysed (Fig. 3, Table 4). Concerning the release kinetics of D-DEX and D-DTX at pH 5, the HPLC and $^1$H DOSY NMR methods obtained almost identical values with 100% of D-DEX released within 5−6 h and maximal D-DTX released within 4 h. The flow-through setting on the surface of the SPR chip accelerated the release of both D-DEX and D-DTX compared to the HPLC and $^1$H DOSY NMR methods, with most D-DEX released within 1 h and 100% of D-DTX within 2 h. The difference in release rate between these methods appears to be directly dependent on the method setup, i.e. a closed system or a flow-through system, the latter being more realistic of in vivo conditions. The D-DEX release kinetics at pH 7.4 obtained by HPLC, $^1$H DOSY NMR were comparable and reached 10–12 % of the released drug within 4 h. On the contrary, the D-DEX release at pH 7.4 determined by SPR was slightly lower, around 6% of the released drug within 4 h. The slower release rate determined by SPR was caused most probably by the SPR method detection limit as the release of D-DEX at pH 7.4 is quite low. The study of D-DTX release at pH 7.4 via HPLC was complicated by the hydrolysis of ester bonds within D-DTX, indicating that all degradation products must be separated and calibrated to be quantified by HPLC. Similarly, no measurable difference in the critical parameter for $^1$H DOSY NMR (D) between the reference and the tested conjugate prevented the quantification of the drug release. Thus, only the SPR analysis could follow the D-DTX release at pH 7.4, showing 20% release after 1 h. Unlike hydrophobic D-DEX and D-DTX, the physicochemical characteristics of HAL enabled its release to be observed by CE and $^1$H NMR, with the results of both methods being in good agreement indicating 20% of released HAL within 4 h at pH 5 and negligible release at pH 7.4. Similarly, the HAL release at pH 5 quantified by SPR analysis was accelerated with approximately 40% of released HAL within 4 h, indicating the importance of a flow system to obtain realistic data. However, negligible HAL release was observed at pH 7.4 in accordance with CE and $^1$H NMR.

In summary, the reproducibility of the methods was sufficiently high, with the flow-through setting leading to faster drug release for all the tested systems compared to the closed system experimental setup. Regarding the final biological use, the SPR method is the most appropriate to obtain representative data about the behaviour of the given system in in vivo conditions, for example, the bloodstream after in vivo application as well as the final destination of the tumour or inflamed tissues. In conclusion, the SPR method is a highly promising method to determine the release kinetics of a wide range of drugs in the most realistic scenario. However, considering that the SPR method also has its limits in sensitivity at low release rates and does not allow the determination of the release rate in a long-term mode, it is advantageous to combine the SPR method with another method that can supplement the data obtained by SPR. As described in this manuscript, the selection of the additional method should be based on the type of the drug molecule studied.

## Conclusion

Herein, we present three approaches utilising SPR, CE and $^1$H DOSY NMR to evaluate the release kinetics of diverse drugs from a polymeric drug delivery system. The methods were optimised and their applicability was evaluated for three biologically active molecules, i.e. anti-inflammatory glucocorticoid dexamethasone (DEX), cytostatics docetaxel (DTX) – both highly hydrophobic and neutral; and the hydrophilic cationic hexyl ester of 5-aminolevulinic acid (HAL), which is a precursor for the biosynthesis of photosensitizer protoporphyrin IX, all bound to the

HPMA-based copolymer via a pH-sensitive hydrazone bond. The overall suitability of the three methods was evaluated using several parameters and compared to the conventional HPLC method, indicating that the SPR method enables the determination of the release kinetics of the widest range of drugs and employs flow-through conditions which are more clinically relevant. Thus, the SPR method is the most promising method to determine the release kinetics in conditions mimicking various environments ranging from the bloodstream after in vivo application to the final tissue destination in the most realistic scenario. Finally, it is advantageous to combine SPR biosensor with another relevant method to obtain unambiguous and reliable drug release kinetics which are crucial for understanding the biological behaviour of the system.

## Materials and methods

The materials, synthesis and the physicochemical characterisation of the prepared copolymeric precursors and conjugates are described in detail in the Supplementary materials and methods. The $^1$H NMR spectra of D-DEX and D-DTX are displayed in the Supplementary Figs. SI1 and SI2. The 1H-NMR spectra of conjugates with D-DEX, D-DTX, HAL and B are displayed in the Supplementary Figs. SI3, SI4, SI5 and SI6 respectively. All the drug release experiments were performed in triplicates, the values are mean of those measurements.

**Drug release experiments - surface plasmon resonance biosensor.** A laboratory SPR sensor based on the wavelength spectroscopy of surface plasmons (Plasmon VI)[40] with six sensing channels and dispersionless microfluidics[41] was used to monitor the drug release from copolymers. In this SPR sensor, the angle of incidence of the light beam is fixed and the SPR dip is observed in the spectrum of polychromatic light coupled to a surface plasmon. The sensor response is expressed in terms of the shift in the wavelength at which the SPR dip occurs and this shift is sensitive to changes in the refractive index caused by the binding of molecules to the sensor surface. A shift of 1 nm in the SPR wavelength represents a change in the protein surface coverage of 17 ng/cm$^2$ [42]. SPR chips were prepared by coating glass substrates with thin layers of titanium (1–2 nm) and gold (48 nm) via e-beam evaporation under a vacuum.

The sensor surface of the SPR chip was initially functionalised with drug-loaded copolymers immobilised to the sensor surface via stable streptavidin-biotin binding to the self-assembled monolayer of mixed carboxy-terminated and hydroxy-terminated oligo-ethylene glycol thiols modified with streptavidin[42]. Briefly, a clean SPR chip was immersed in an ethanolic solution of HS-OEG-COOH and HS-OEG-OH thiols (c$_{total}$ = 200 μM, molar ratio 3:7) for 10 min at 40 °C and stored at room temperature for at least 12 h. Then, the chip was rinsed with ethanol and Q-water before being mounted into the SPR sensor. An aqueous mixture of 12.5 mM NHS and 47.6 mM EDC was injected for 10 min to activate the carboxylic groups. A solution of streptavidin in SA$_{10}$ (50 μg/ml) was pumped along the sensing surface for 15 min to allow streptavidin to covalently bind to the activated carboxylic groups. Then, short (5 min) injections of PBS$_{NaCl}$ and 0.5 M aqueous ethanolamine were used to remove the non-covalently bound streptavidin and to deactivate the remaining carboxyl groups, respectively. Finally, copolymers with and without drugs dissolved in PBS were injected into channels to allow the biotinylated copolymers to bind to the immobilised streptavidin until the sensor responses achieved ~3 nm corresponding to the copolymer surface densities of ~50 ng/cm$^2$. Then, the solution of biotin 4-amidobenzoic acid sodium salt dissolved in PBS (10 μg/ml) was injected onto a sensor surface to bind to

unoccupied streptavidin pockets to prevent further binding of biotinylated copolymers to the immobilised streptavidin.

Since the SPR sensor response in the detection channel is also sensitive to the interfering effects (thermal drifts, the reaction of the copolymer to the change of pH, etc.), the sensor surface modified with copolymers without drugs (reference channel) was prepared under the conditions described above and used to compensate these effects. To obtain the sensor response associated only with the drug release, the reference-compensated sensor response was determined by subtracting the sensor response in the reference channel from that in the detection channel.

The drug release was triggered after the functionalisation of a sensor surface with drug-loaded copolymers. Buffers of PBS (pH 7.4), MES$_{NaCl}$ (pH 5), and CB$_{NaCl}$ (pH 3) were injected for several hours to the detection and reference channels. When the pH-sensitive linker was cleaved, the drug was released from the immobilised copolymer and washed from the sensor surface by the buffer solutions producing a shift in the wavelength of the SPR dip to the blue wavelength region (decrease of a surface density of immobilised molecules). In all SPR biosensing experiments, the volumetric flow rate and temperature were kept at 20 µl/min and 37 °C, respectively. The curves obtained were averaged from at least three experiments and the grey areas denote the release standard deviations.

**Drug release experiments - capillary electrophoresis**. Monitoring of the released HAL was performed on an Agilent 7100 CE system (Agilent Technologies, Waldbronn, Germany). The C$^4$D detector (Admet, Czech Republic) used for capacitively coupled contactless conductivity detection consisted of two tubular electrodes, 4 mm long with a 1-mm insulation gap and the inner diameter of the electrodes was 400 µm. The detector was operated at a frequency of 1.84 MHz with an amplitude of 44 V. Unmodified fused silica capillary, 20 µm i.d., 375 µm o.d., 50.0 cm total, and 35.0 cm effective length (Polymicro Technologies, Phoenix, USA) was thermostatted at 25 °C. Samples were injected using a pressure of 5 kPa for 20 s and the separation voltage was 25 kV (current approx. 7 µA). A pressure of 10 kPa was applied to the inlet end of the capillary during the analysis to suppress the negative effects of the sample matrix, i.e. to improve the repeatability of migration times and to stabilise the baseline signal of the detector and 1 M formic acid was used as a background electrolyte. Before each run, the capillary was rinsed for 3 min with 1 M NaOH, 2 min with water and 4 min with background electrolyte using a pressure of approximately 93 kPa. OpenLab software was used for data acquisition and analysis and the migration times of HAL and Tris–HCl (internal standard) were 4.3 and 3.6 min, respectively.

P3-HAL (5 mg/ml) dissolved in phosphate-citrate buffers of appropriate pH was incubated at 37 °C. After 2, 5, 9, and 24 h, 40 µl of the polymer solution was pipetted into a CE sample vial and 4 µl of Tris–HCl solution (5 mg/ml) was added as an internal standard. The sample was injected into the CE system without any further treatment. All samples were prepared and measured in triplicates.

Calibration standards were prepared in concentrations of 0.02, 0.05, 0.10, 0.20, 0.50 and 1.00 mg/ml of HAL and contained 5 mg/ml polymer precursor (P3) to match the standard matrix with the matrix of measured samples. All calibration samples contained Tris–HCl as an internal standard and the calibration curve was constructed by plotting the ratio of HAL and Tris–HCl peak areas as a function of HAL concentration. A quality control sample was prepared at the 0.50 mg/ml concentration of HAL and was measured after every 3 runs. The amount of HAL released from the polymer was expressed

as the percentage of HAL released from the polymer kept in 50 mM HCl at 37 °C overnight.

**Drug release experiments - $^1$H NMR spectroscopy**. $^1$H NMR spectroscopy was applied to study the HAL release rate from P3-HAL. The time dependences of the $^1$H NMR spectra were measured at 37 °C at pH 5 and 7.4 and for quantification, the changes in signal intensity at $\delta \approx 4.2$ and 2.95 ppm in time were used. These signals "1 + 4 and 2" correspond to attached and free HAL respectively (see Supplementary Figs. SI5 and SI12).

$^1$H DOSY NMR spectroscopy was used to study the drug release from P1-D-DEX, P2-D-DTX and P3-HAL using an NMR Bruker Avance III spectrometer operating at 600 MHz ($^1$H) with D$_2$O based buffer (pH 5 and pH 7.4) at 37 °C. Time dependence $^1$H diffusion experiments were performed using a DiffBB probe head with 40 A gradient amplifiers and the gradient strength was varied in 16 steps. A Pulsed Field Gradient Double-Stimulated Echo sequence was used to correct any convection effects. In all cases, the pulse gradient time $\delta$ was 1 ms, diffusion time $\Delta$ 20 ms and maximum gradient value = 5.69 T/m. From each experiment, the integrated intensities (I) as a function of the applied gradient (g) were obtained and then the diffusion coefficients (D) were computed using single exponential decay by fitting the Stejskal–Tanner equation[43]:

$$I = I_0 \exp[-Dg^2\gamma^2\delta^2(\Delta - \delta/3)] \qquad (1)$$

The diffusion data were processed with Dynamics Centre software and the D values of polymer precursors and conjugates were monitored at preselected time points. The polymer precursors P1 and P3 were used as a reference, see representative Supplementary Fig. SI10, where the changes of D of P1-D-DEX and P1 are displayed. The final drug release rates of D-DEX and D-DTX were calculated from D changes of conjugates in time subtracting the D values of precursor from respective numbers of conjugates. The D values were fitted for quantitative characterisation at all time points. For D-DEX and D-DTX, the D was expressed as an average of the three signals from the HPMA comonomeric unit (-CH, -NCH- and -CH$_3$ marked as e, d and b + f respectively displayed on the structure in Supplementary Fig. SI3) and fitted with single component fitting. Regarding HAL samples, some of the signals of free and attached HAL overlapped ("1 + 4" at $\delta \approx 4.2$ ppm and "3" at $\delta \approx 2.78$ ppm, see Supplementary Figs. SI11 and SI12), therefore, the D values were additionally fitted with two-component fitting.

**Reporting summary**. Further information on research design is available in the Nature Portfolio Reporting Summary linked to this article.

## Data availability

The data that support the findings of this study are available from the corresponding author upon reasonable request.

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

## Acknowledgements

This work was supported by the Ministry of the Health of the Czech Republic (NU20-08-00255), the Czech Science Foundation (grant 19-00956Y), and the project National Institute for Cancer Research (Programme EXCELES, ID Project No. LX22NPO5102) funded by the European Union, Next Generation EU, and Charles University (project SVV260690).

## Author contributions

A.L. – conceptualisation, methodology, synthesis, HPLC analysis, writing - original draft; T.Š. – SPR analysis, writing - review & editing; L.P. – SPR analysis; K.K. – synthesis; R.K. – ¹H NMR and ¹H DOSY NMR analysis; A.Š. – CE analysis; T.K. – CE analysis, funding, writing - review & editing; J.H. – conceptualisation, consultation; E.R. - conceptualisation, methodology, writing - original draft, writing - review & editing; T.E. – conceptualisation, supervision, writing - review & editing.

## Competing interests

The authors declare no competing interests.
