## [Peer Review File · Communications Chemistry]

Reviewers' comments:

Reviewer #1 (Remarks to the Author):

The authors reported three novel approaches utilising SPR, CE and ¹H DOSY NMR to evaluate the release kinetics of diverse drugs from a polymeric drug delivery system. They used several parameters and compared to the conventional HPLC method to evaluate the overall suitability of these three methods, indicating that the SPR method enables the determination of the release kinetics of the widest range of drugs. The manuscript shows detailed method and it is well organized. I believe this work is suitable for publication in Communications Chemistry, but there are a few points that should probably be addressed more completely.

1. The author should add more discussions on the advantages and deficiencies of these three methods, especially compared to the commonly used test methods.
2. The author describes that SPR method is a highly promising method to detect drug molecules, whether it has an advantage in terms of simple operation.
3. Figures in the manuscript are more vague, higher definition figures should be provided.

Reviewer #2 (Remarks to the Author):

In this manuscript, the authors optimized and applied three different methods (SPR, CE, and NMR) to determine the pH-triggered release of three different polymer nanomedicines. Different parameters were measured, and the results were compared between methods and with the golden standard method of high-performance liquid chromatography. Overall, it is a well-designed and well-written manuscript, and the findings in this paper are interesting and helpful to the field. However, there are some concerns and details that are suggested to be addressed before publication.

My specific comments are listed below:

- 1) Can the authors provide more rationale for choosing pH-triggered drug release? Will other aspects such as temperature, site microenvironment, and mechanical regulation affect the findings here/ or are they worth investigating?
- 2) Have the authors considered the relationships between the drug release, the half-life time of the drug, and the drug's therapeutic effectivity? In other words, is it worth considering the drug's half-life time when we evaluate the drug release kinetics?
- 3) It may be better to describe the standard/rationale for why these three drugs were chosen.
- 4) It seems like the SPR had slightly different preparation procedures. If so, please comment if this will affect the findings/results or not?
- 5) pH 5 and 7.4 were measured to represent the lysosomes and blood conditions, respectively. Have the authors tested any pH values between these two?
- 6) It seems like there was a discrepancy between the SPR and HPLC D-DEX-7.4 (Fig 3 A-1 and A-

3). If so, please comment.

7) Section "Release kinetics determined by SPR biosensor", "However, pre-analysis modification of the sample is crucial", would be good if the author provided more details.

8) In the conclusion, the authors claimed that "it is advantageous to combine SPR biosensor with another relevant method to obtain unambiguous and reliable drug release kinetics", it would be better to put more descriptions to describe and support this statement.

9) Have the author used FTIR to compare the synthesis polymers?

Minor:

1) 1st sentence of the last paragraph of the introduction, it may be better to use ^1H DOSY NMR or NMR rather than DOSY.

2) Maybe it is better to spell out HPMA when you first mention it.

3) Suggest to improve the resolution of the figures.

4) Any replicated experiment? Are the results here from a single drug single measurement?

5) It would be better if the authors could provide the experimental setup diagram(s) for the material and methods section.

Responses to reviewers' comments:

Reviewer 1:

1. The author should add more discussions on the advantages and deficiencies of these three methods, especially compared to the commonly used test methods.

Response: *We thank the reviewer for this particular comment. To make the discussion clear and understandable for the readers, we arranged within the results and discussion part a special chapter "Head-to-head comparison of the methods used for release kinetics", in which the discussion of pro and cons of all the methods is carried out. There are several subchapters dealing with different aspects of the drug release analysis. We believe that we have carefully discussed all the obtained results.*

2. The author describes that SPR method is a highly promising method to detect drug molecules, whether it has an advantage in terms of simple operation.

Response: *In the manuscript we have described the SPR method as a quite universal method to determine the rate of the release of various drugs from the polymer drug delivery systems. Thus, it enables to measure both the hydrophilic and hydrophobic drugs, charged drugs and drugs which do not have any significant UV-Vis absorption, nor fluorescence. As mentioned, the method has an advantage in terms of simple operation, however, it can offer much more benefits like low sample consumption, continuous measurement of drug release, drug versatility etc. It is only necessary to enrich the polymer system with biotin molecule, which is needed for the SPR chip attachment.*

3. Figures in the manuscript are more vague, higher definition figures should be provided.

Response: *We thank the reviewer for this comment. The figures are now provided in better resolution. We attach also the separated files of figures.*

Reviewer 2:

1. Can the authors provide more rationale for choosing pH-triggered drug release? Will other aspects such as temperature, site microenvironment, and mechanical regulation affect the findings here/ or are they worth investigating?

Response: *In general, there are several mechanisms of stimuli responsive drug delivery systems. In particular for polymer-based drug delivery systems with covalently bound drugs, the most common mechanisms of drug activation are mediated by pH-sensitive, enzymatically and reductively cleavable linkers. All three are based on the tumor microenvironment conditions: either using the decreased pH within the tumor or the pH of the lysosome of the cancer cells; employing the substrate specificity of the enzymes localized in higher extent on the tumorous tissue or in lysosomes of the tumor cells; or finely the reductive conditions in the cancer cells given by the glutathione and thioredoxin I level. Here, we have selected pH-triggered drug release as one of above-mentioned mechanisms, since the pH-responsive materials have been used not only in cancer therapy but also in the therapy of inflammatory diseases. Moreover, in order to precisely compare the tested methods and reduce other interfering*

effects, the pH-responsive polymer conjugates were selected since the setup of release experiments requires only the change of the pH value. In the case of enzymatic or reductive degradation we believe that the experimental setting and the characteristics of the drugs differ so broadly that it would be a limiting factor for our manuscript.

The pH of the target site plays a crucial role in determining the drug release behavior. The pH microenvironment in solid tumors is known to be acidic compared to healthy tissues based on the Warburg effect. Moreover, the pH is even more decreased in the endosomes and lysosomes of the tumor cells. pH-triggered release provides a means to enhance therapeutic efficacy while minimizing off-target effects and reducing systemic toxicity.

Regarding the other aspects mentioned. Concerning the temperature, the measurements were performed at 37°C to be most similar to the temperature of human body. There are also systems which can release the drug once the temperature is elevated. In this study such systems were not involved, but the SPR method could be advantageously used here also for the drug release rate determination. Importantly, site microenvironment plays a significant role as mentioned above, as the pH is decreased and the specific enzymes can be used for the drug release as well. We believe that the SPR method described in the manuscript could be used also in other drug release systems after solving some optimization issues.

2. Have the authors considered the relationships between the drug release, the half-life time of the drug, and the drug's therapeutic effectivity? In other words, is it worth considering the drug's half-life time when we evaluate the drug release kinetics?

Response: We fully agree that there is a direct relationship between drug release, drug half-life in the bloodstream, and drug therapeutic efficacy. The half-life of a drug in the bloodstream is a critical parameter that can significantly affect the therapeutic efficacy of the administered drug. We believe that release rate is one of the critical parameters that affects the half-life of a drug in the bloodstream. It is important that the drug transport system has sufficient stability in the bloodstream and thus ensures prolonged circulation of the drug in an inactive form protected from biodegradation. On the contrary, it is important that the drug is activated after entering the tumor or inflamed tissue at a speed sufficient for the desired therapeutic effect. One of the advantages of our HPMA-based drug delivery systems is the prolongation of the half-life of the drug in the bloodstream. Knowing the optimal drug circulation half-life, it is then possible to design and adjust the drug-carrying system to release the drug at the desired rate. However, it is also important to mention that drug circulation half-life is not only a matter of release rate under model conditions, there are many other substances in the bloodstream that can also interact with the drug itself, even if it is bound to a polymer carrier. Therefore, these factors must also be considered.

3. It may be better to describe the standard/rationale for why these three drugs were chosen.

Response: We thank the reviewer for this important comment. We have added the rationale for the drug selection at the end of the introduction part. In general, we aimed to provide a diverse set of drugs that represent different therapeutic areas and the selection was based on several factors, mainly their structure and physicochemical properties. First drug, dexamethasone, was chosen because it is a widely prescribed medication for many inflammatory diseases, however it has serious side effects affecting the healthy tissue especially after long-term exposure. Together with high hydrophobicity, it is a perfect candidate for improvement of biological properties via drug delivery systems. This also applies to the second drug, docetaxel which is a chemotherapeutic agent used in the treatment of various cancers

(including breast, lung, prostate, gastric and many more). In terms of physico-chemical properties, docetaxel has another disadvantage (except hydrophobicity) which is the instability of the structure caused by presence of hydrolysable ester bonds causing complications during drug release measurement by HPLC, NMR etc. Last but not least, hexyl ester of amino levulinic acid, which is currently in clinical trials as a very promising tool for photodynamic therapy of cancer, was selected as a hydrophilic and charged molecule with no UV/VIS signal making the analysis of the drug release even harder.

4. It seems like the SPR had slightly different preparation procedures. If so, please comment if this will affect the findings/results or not?

Response: We thank the reviewer for this important comment. The SPR method requires the biotin bound to the polymer as an anchor for the SPR chip attachment. However, this does not affect the drug release rates, the addition of the biotin molecule did not change the characteristics of the polymer conjugates and did not change the characteristics of the buffers used in the experiments. The flow-through setup of the SPR method is importantly closer to the real situation in the blood stream or in final tissues, than the experiment performed in the closed tube. Thus, we believe that the SPR procedure is more realistic for determination of the drug release kinetics.

5. pH 5 and 7.4 were measured to represent the lysosomes and blood conditions, respectively. Have the authors tested any pH values between these two?

Response: The selection of pH 5 and 7.4 was based on established physiological pH values typically associated with lysosomal compartments and blood. These pH values were chosen to provide a comparative analysis of drug release behavior under conditions that mimic the relevant biological environments. Because of the number of the samples and number of methods used, the other pH values were not studied. Anyway, we believe that the SPR is suitable also for other pH as pH 6.0 modeling the endosomes of the cells, pH 6.5 – 6.8 modelling the extracellular environment of the solid tumors and inflamed tissues.

6. It seems like there was a discrepancy between the SPR and HPLC D-DEX-7.4 (Fig 3 A-1 and A-3). If so, please comment.

Response: We thank the reviewer for this important comment. The release of D-DEX at pH 7.4 determined by SPR was a little bit lower during the 4 h of the measurement compared to the results obtained by the HPLC method. We believe that this small discrepancy between the methods is given by the SPR method sensitivity as the release of D-DEX at pH 7.4 is quite low. We have added explanation into the manuscript.

7. Section "Release kinetics determined by SPR biosensor", "However, pre-analysis modification of the sample is crucial", would be good if the author provided more details.

Response: We thank you for this important comment. We have added several sentences to describe this issue in more detail.

8. In the conclusion, the authors claimed that “it is advantageous to combine SPR biosensor with another relevant method to obtain unambiguous and reliable drug release kinetics”, it would be better to put more descriptions to describe and support this statement.

Response: *We agree on this particular comment with the reviewer and we have added description into the manuscript in the last section of results and discussion. The SPR was found as the most powerful method for determination of drug release, nevertheless, the SPR method has its limitations in terms of sensitivity of slow release rates and is not suitable for long term determination of drug release. Thus, to reach complete release rate data it is advantageous to combine the SPR method with the second method given supporting data for those data obtained by the SPR.*

9. Have the author used FTIR to compare the synthesis polymers?

Response: *We did not use FTIR to compare the synthesized polymers. Polymers were fully characterized via different methods, e.g., GPC, DLS, ^1H NMR. The functional groups present on the polymers were characterized via UV/VIS spectroscopy.*

Minor:

1st sentence of the last paragraph of the introduction, it may be better to use ^1H DOSY NMR or NMR rather than DOSY.

Response: *The sentence was corrected.*

Maybe it is better to spell out HPMA when you first mention it.

Response: *Thank you for this important point. The HPMA was described when first used.*

Suggest to improve the resolution of the figures.

Response: *The resolution of the figures was improved.*

Any replicated experiment? Are the results here from a single drug single measurement?

Response: *All the values are mean of 3 independent measurements. The information was added into the material and method part of the manuscript.*

It would be better if the authors could provide the experimental setup diagram(s) for the material and methods section.

Response: *Within the Figure 4 we tried to describe in detail the pros and cons of the methods studied. Also, all the methods were described in detail to give the readers the possibility to follow up the research methodology. Especially the SPR method was described in detail with the aim to be used by other colleagues for the release rate determination. Thus, we believe that the experiment setup is now well described and easy to follow.*

REVIEWERS' COMMENTS:

Reviewer #1 (Remarks to the Author):

The revised manuscript is acceptable for publication.

Reviewer #2 (Remarks to the Author):

The authors have addressed all of my comments and concerns, and I do not have further comments and suggestions. Thanks